# Analysis of the Holarctic *Dictyoptera aurora* Complex (Coleoptera, Lycidae) Reveals Hidden Diversity and Geographic Structure in Müllerian Mimicry Ring

**DOI:** 10.3390/insects13090817

**Published:** 2022-09-07

**Authors:** Michal Motyka, Dominik Kusy, Renata Bilkova, Ladislav Bocak

**Affiliations:** Czech Advanced Science and Technology Institute, Palacký University, Slechtitelu 27, 779 00 Olomouc, Czech Republic

**Keywords:** taxonomy, cryptic species, mtDNA, rRNA, barcode, dispersal, last glacial maximum, aposematic signal, Müllerian mimicry, morphological stasis

## Abstract

**Simple Summary:**

We evaluated the red net-winged beetle populations’ morphological and genetic divergence within the Holarctic region. In contrast with relatives, *D. aurora* occurs in an exceptionally large range and very different ecosystems. In Northern America, we found an earlier undetected cryptic species isolated by the Bering Strait since the mid-Miocene. *D. aurora* colonized Fennoscandia from at least two refugia after the last glacial maximum. The absence of morphological differentiation is supposedly affected by the selection for similarity in the Müllerian mimicry ring. The results exemplify the phylogeographic history in the Holarctic region and contribute to understanding the morphological stasis in an extensive circumpolar range.

**Abstract:**

The elateroid family Lycidae is known for limited dispersal propensity and high species-level endemism. The red net-winged beetle, *Dictyoptera aurora* (Herbst, 1874), differs from all relatives by the range comprising almost the entire Holarctic region. Based on a five-marker phylogeny and 67 barcode entries (*cox1-5*′ mtDNA) from the whole range, we recovered two genetically distinct species within traditionally defined *D. aurora* and resurrected the name *D. coccinata* (Say, 1835) as the oldest available synonym for Nearctic populations. Yet, no reliable morphological trait distinguishes these species except for minute differences in the male genitalia. *D. coccinata* is a monophylum resulting from a single Miocene dispersal event, ~15.8 million years ago, and genetic divergence implies long-term isolation by the Bering Strait. Far East Asian and west European populations are also genetically distinct, although to a lower extent. Two independent colonization events established the Fennoscandian populations after the last glacial maximum. Besides intrinsic factors, the high morphological similarity might result from stabilizing selection for shared aposematic signals. The rapidly accumulating barcode data provide valuable information on the evolutionary history and the origins of regional faunas.

## 1. Introduction

Molecular data have become essential sources of information in biodiversity studies, and they are especially valuable if diagnostic morphological traits are contentious [1,2,3,4,5]. Then, DNA data can solve the delimitation of cryptic species [2,6,7]. Mitochondrial markers are also widely used for the rapid genetic screening of biodiversity, and the method can be increasingly used with next-generation sequencing [8,9,10,11]. Although the mtDNA resolution to detect subtle intraspecific differentiation is low [7,12,13,14], even short fragments are sufficient for obtaining the first clue to shallow relationships and identification of discrepancies between traditional morphology-based taxonomy and genetic diversity [2,15,16].

Here, we focus on an exceptionally widespread red net-winged beetle, *Dictyoptera aurora* (Herbst, 1784) (Erotinae: Dictyopterini) that belongs to the predominantly tropical family Lycidae [17,18]. Unlike relatives, *D. aurora* has been reported from almost the entire Holarctic region. Hence, it is a striking case of a single net-winged beetle with circumpolar distribution, spanning four continents and very different ecosystems [19,20,21]. Although the species occurring in high-latitude regions are usually widespread (Rapoport’s rule; [22,23], only a few Palearctic lycids distantly resemble the distribution of *D. aurora*. *Lygistopterus sanguineus* (L.) and *Pyropterus nigroruber* (De Geer) occur in Eurasia yet are absent in the Nearctic region. *Platycis minuta* (Thomson) is known from Europe and only marginally reaches Western Siberia [21]. Similarly, few lycids have a wide distribution in the Nearctic region, and most occur in the continent’s eastern or western regions [19,20,24,25,26].

Low gene flow supposedly enables small-scale diversification, which has been well documented by recent wide-scale diversity analyses of net-winged beetles [27,28,29,30]. The observed distribution patterns can at least partly be explained by lycid biology. The larvae depend on permanently moist rotten wood as a source for larval feeding and shelter [20,31]. Additionally, adults live shortly, mostly sit on leaves and dead wood, and do not fly outside the forest canopy. Due to the low vagility, their ranges are fragmented by the presence of semiarid areas, steppes, deserts, and even seasonally dry lowlands that harbor a much lower number of species [28,32]. For example, the southern boundary of net-winged beetle distribution in the Holarctic is strictly limited by the steppe and desert regions in Northern Africa, Central Asia, the southwestern USA, and Mexico [17,19,24,25,26]. Similarly, the eastern and western populations of *D. aurora* are separated by unfavorable conditions in the wide Mississippi valley.

Within the Holarctic, the ranges were heavily influenced by the Late Tertiary and Quaternary climatic oscillations [33,34,35]. As evidence, we can point to the absence of *L. sanguineus* in the British Isles in contrast with its wide distribution from the French Atlantic coast to Sakhalin [21,36]. As a dominantly tropical group, the diversity of Lycidae decreases rapidly with the latitude [17,37,38]. *D. aurora* is one of the few species tolerant to very different climatic conditions, including low winter temperatures. This species occurs from the northern limits of forest habitats in northern Fennoscandia, Canada, and Alaska to Northern Africa, and Arizona [19,20,21].

The present study surveyed the genetic structure of the *D. aurora* complex and compares the results with other widespread net-winged beetles of the Palearctic region. We will detail the morphology to identify phenotypic differences between geographically distant populations with known genetic divergence. Based on previous studies, we expect low genetic homogeneity across geographical regions [28,29,39,40]. Further, we will discuss the genetic diversity of *D. aurora* in the recently colonized areas of Europe that have been available only since the ice sheet retreated, which covered most of Northern Europe during the last glacial maximum (LGM; [35]). The striking morphological similarity of geographically distant populations of the *D. aurora* complex questions putative drivers affecting phenotypic diversification recently proposed by Ferreira et al. [41].

## 2. Materials and Methods

### 2.1. Material, DNA Data and Phylogenetic Analyses

The morphospecies *D. aurora* was collected in the Czech Republic, Croatia, Russian Far East, and the USA, and sequenced for the D2 loop of *LSU* rRNA, *SSU* rRNA, *rrnL*, *nad5*, and *cox1-3*′ mtDNA fragments. Considering the discovered deep split between Nearctic and Palearctic populations, we further separately analyzed the *cox1-5*′ (barcode) mtDNA fragment downloaded from the BOLD database (http://www.boldsystems.org, accessed on 1 June 2022) as we did not have properly fixed individuals of *D. aurora* from many regions. We also analyzed barcodes of *P. nigroruber*, *Pl. minuta*, and *L. sanguineus* occurring in Europe to compare intraspecific genetic diversity.

The nuclear rRNAs, *rrnL* mtDNA, and tRNAs were aligned using MAFFT 7.2 with the Q-INS-I algorithm [42]. Protein-coding genes were aligned using Trans-Align [43] and eye-checked for stop codons. The fragments were concatenated and analyzed under the maximum likelihood (ML) criterion in IQ-TREE v.1.6 [44]. The model selection for each gene was performed with ModelFinder [45,46] implemented in IQ-TREE2 (-MFP option). The GTR model was applied in all analyses. Ultrafast bootstrap [47] values were calculated in IQ-TREE (options -bb 5000) to assess nodal supports for focal relationships.

Further, we generated a haplotype network based on the *cox1-5*′ fragment using the TCS algorithm [48] in PopART [49]. The output was processed in Adobe Illustrator.

The dated tree of the genus *Dictyoptera* Latreille was taken from the whole family analysis (the dataset used by [50] and expanded by unpublished data). Beast v.1.8 [51] was used to date splits among the Lycidae family with calibrated separation of Lycidae + *Iberobaenia* to 176 my. The following settings were employed: the unlinked partitions (genes and codon positions were partitioned in the (1 + 2),3 scheme, the HKY + I+G model, and the birth–death speciation prior. The run was computed for 5 × 10^7^ generations with a sampling frequency of 10,000. The ESS values and plateau phases were checked using TRACER v.1.681 [52]). The maximum credibility tree was generated with TREEANNOTATOR v.1.8.1 [51].

### 2.2. Morphological Study

About 100 individuals deposited in the senior author’s laboratory from the Nearctic and Palearctic regions were investigated to identify putative diagnostic characters and intraspecific variability. We specifically compared the shapes of the antennae, pronotum, and elytral costae. Further, we dissected multiple individuals from the whole range to study the intraspecific variability in the male genitalia. The voucher specimens were relaxed in 50% ethyl alcohol for an hour and then detached abdomens were treated with a hot 10% KOH solution to remove muscles and body fat. The external characters and genital morphology were observed under an Olympus SZX-16 microscope and photographed by an attached digital camera. We used Helicon Focus (www.heliconsoft.com, accessed on 21 January 2022) and Photoshop 13.0 (Adobe Inc., San Jose, Ca, USA) to assemble photo stacks in figures showing the principal morphological traits.

The individuals used for morphological investigation and sequenced specimens bearing the UPOL voucher number are deposited in the collection of the Laboratory of Biodiversity and Molecular Evolution, CATRIN-CRH, Olomouc.

## 3. Results

### 3.1. Phylogeny, Genetic Diversity and Distribution

In our five-gene phylogenetic analysis, we find two distant clades represented by the individuals traditionally assigned to the morphospecies *D. aurora* (Figure 1). The North American populations are represented by a single individual from the Eastern USA, and the Euro-Asian populations by three individuals from the Russian Far East and four from Central Europe (Figure 1C). The dating analysis using the *cox1-3*′, *rrnL*, *nad5* mtDNA dataset suggests that Nearctic and European populations separated already in the mid-Miocene, 15.75 million years ago (Figure 1D). Their generic divergence is documented by uncorrected pairwise distances in mitochondrial and nuclear markers (complete *SSU* RNA 0.3%, D2 loop of *LSU* rRNA 0.3%, *rrnL* mtDNA 4.3%, *cox1-3*′ 13.8%, *nad5* 9.6%). We suggest the presence of *D. aurora* in the Western Palearctic shortly before the end of the Miocene, 6.23 mya when the Far East and European populations split (Figure 1D).

Further, the *cox1-5*′ mtDNA (barcode) fragments were analyzed from the whole range. The uncorrected pairwise distances of the barcode *cox1-5*′ fragments reached 12.5% (Appendix A). The phylogenetic network based on the *cox1-5*′ fragment confirms separate clusters of haplotypes corresponding to the main branches recovered by the phylogenetic analysis (Figure 1B,C,E). Further, we identified European populations’ relatively high intraspecific genetic diversities. The genetic divergence between Fennoscandian populations reaches 7.2% in *cox1-5*′ (barcode) fragment. Additionally, we constructed the barcode networks for three lycid species occurring in Central Europe and Fennoscandia (Figure 1F–H). *L. sanguineus* and *Pl. minuta* comprise genetically close populations (Figure 1F,G). *P. nigroruber* contains genetically divergent populations in Fennoscandia (Figure 1H).

**Figure 1 insects-13-00817-f001:**
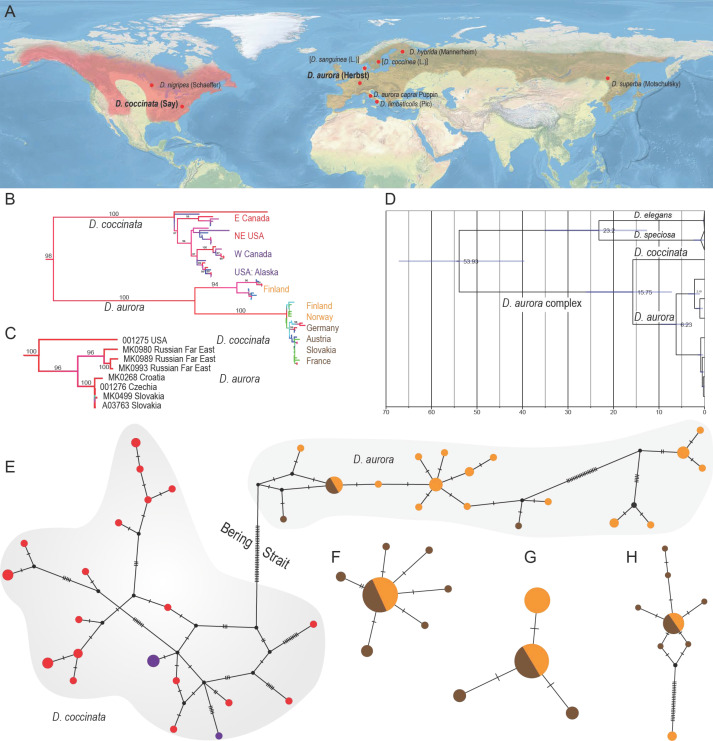
(**A**) Distribution of *Dictyoptera aurora* (Herbst) and *D. coccinata* (Say); the red dots designate type localities of individual taxa proposed in the *D. aurora* complex; valid names are given in bold, available names (younger synonyms) in regular font and unavailable names (junior primary homonyms) in square brackets. (**B**) The *cox1-5*′ mtDNA tree as recovered by the IQ-tree maximum likelihood (ML) analysis. The node labels show bootstrap fractions. Shallow relationships are not statistically supported, and bootstrap fractions are omitted. (**C**) ML tree of the *Dictyoptera aurora* complex inferred from rRNA and mtDNA markers within the analysis of the whole family. (**D**) Dated tree of three species of *Dictyoptera*, the part of the whole-family tree displayed. (**E**–**G**) Haplotype networks constructed from barcodes: (**E**) *D. coccinata* and *D. aurora*; (**F**) *Lygistopterus sanguineus* (L.); (**G**) *Platycis minuta* (Thomson); (**H**) *Pyropterus nigroruber* (De Geer). Pie colors designate the following regions (corresponding to label colors in (**B**)): red—Eastern USA and Canada; violet—Alaska and Western Canada; orange—Fennoscandia; brown—the rest of Europe; black pies designate hypothetical haplotypes.

### 3.2. Taxonomy

Family Lycidae**Tribe Dictyopterini Houlbert, 1922**Dictyopterini Houlbert, 1922: 338. [53]
***Dictyoptera* Latreille, 1829***Dictyoptera* Latreille, 1829: 464 [54].Type species *Cantharis sanguineus sensu* Linnaeus, 1761 (=*Pyrochroa aurora* Herbst, 1784).*=Dictyopterus* Mulsant, 1838: 80 [55].Type species *Pyrochroa aurora* Herbst, 1784.
***Dictyoptera coccinata* (Say, 1835), reinstated name***Omalisus coccinatus* Say, 1835: 155 ([56]; Type locality USA: Pennsylvania, Indiana; the syntypes destroyed).*=Eros nigripes* Schaeffer, 1911: 121 ([57]; Type locality USA: Minnesota).

#### 3.2.1. Differential Diagnosis

Both *D. coccinata* and *D. aurora* are morphologically highly variable species, and we have not found any external traits for identifying these sister species (Figure 2 and Figure 3). We only noted minor, potentially unreliable, differences in the shape of the phallus. The basal process (pointing ventrally to the phallobase) is slenderer in *D. coccinata.* Still, there are intraspecific differences, and although the trait is very apparent in some individuals (Figure 3M), it is hardly observable in others (Figure 3O). Additionally, the ventral edge of the *D. coccinata* phallus is concave to straight in the basal half, unlike the simply concave edge in the whole length of *D. aurora* (observed in the ventrolateral view; Figure 3G–Q).

#### 3.2.2. Intraspecific Variability

High intraspecific variability of external characters includes the shape of the male antennae, pronotum, and the arrangement of elytral costae (Figure 2A–J). The males have slenderer antennae than females, but we noted differences in the relative length of antennomeres also in males (Figure 3A–E). Similarly, we observed differences in the shape of the pronotum (Figure 2E–J). Unlike fully sclerotized beetles, the asymmetry of the pronotum is commonly encountered, and defective structures are common in soft-bodied lycids (Figure 2F). The net-winged beetles are so weakly sclerotized that dry-mounted individuals have regularly deformed elytra. Then, it is difficult to compare the strength of elytral costae. Therefore, we photographed the specimens immediately after mounting to show the differences (Figure 2A–D). The specimens in Figure 2A,C,D have slightly stronger primary costa 2 that reaches the apex of elytra; the specimen in Figure 2B has merged primary costae 2 and 3. The primary costa 2 is not only shorter but also relatively weaker. Similarly, the male genitalia is variable. There are differences in the basal part of parameres (pointed in Figure 3L,M and straight in Figure 3P,Q), the conspicuousness of thorns at the inner margin of parameres (acute in Figure 3L,M,G, obtuse in Figure 3P,Q), and the presence of median keel in the phallobase (Figure 3L–Q).

#### 3.2.3. Biology

*D. coccinata* is a locally common forest species in mid to high-latitude mountains; its larva is unknown. Adults are active from March in the south until late June in the high mountains and the northern part of its range.

#### 3.2.4. Distribution

Nearctic Region: USA (from northern Florida and Eastern Texas to the Canadian border; Arizona, New Mexico, eastern slopes of the Rockies, southwestern South Dakota, Alaska) and Canada to the northern limit of spruce forests. Most records are available from the Rockies and Appalachian Mountains [19]. Further information was taken from www.inaturalist.org, accessed on 2 July 2022. The records close to range limits were revised, and misidentifications were excluded). The northernmost record is known from central Alaska (Chena River, 65.9° N).


**
*Dictyoptera aurora (Herbst, 1784)*
**
*Pyrochroa aurora* Herbst, 1784: 105 [58] (Type locality Germany).=*Lampyris coccinea* Linnaeus, 1767: 646 [59] (Type locality Sweden?).=*Dictyopterus hybridus* Mannerheim, 1843: 88 [60] (Type locality Finland).=*Dictyoptera aurora limbaticollis* Pic, 1914: 50 [61] (Type locality Italy).=*Cantharis sanguineus sensu* Linnaeus, 1761: 202 [62] (Type locality Denmark)=*Dictyopterus superbus* Motschulsky, 1860: 115 [63] (Type locality Russian Far East: Amur).=*Dictyoptera aurora caprai* Pupin, 1974: 40 [64] (Type locality Italy).*Differential diagnosis*. See *D. coccinata* above. The species is morphologically variable (Figure 2 and Figure 3).

#### 3.2.5. Description of Second Instar Larva

Body—slender, cylindrical, slightly tapering anteriorly and posteriorly. Sclerites—light brown, membranes—yellowish white.

Head narrower than the frontal edge of pronotum, transverse, slightly depressed (Figure 4A,B). The dorsal part of epicranium plate-like, very slightly projected anteriorly in middle of the frontal edge, lateral part of head membranous (Figure 4H). Blind, ommatidia absent. Antennae two-segmented, basal antennomere transverse, very short; apical antennomere cylindrical, with membranous apex, bearing dorsal, peg-like process (Figure 4G,H). Mandibles are relatively short, robust, slightly curved, bases close, mandibles widely divergent, and incapable of biting or chewing (Figure 4E,F). Palpifer segment-like, transverse, shorter than maxillary palpi. Mala—tiny, membranous, with short setae. Maxillary palpi three-segmented, with basal palpomeres transverse, middle palpomere slightly conical, apical palpomere slender, long (Figure 4E,H). Prementum transverse, labial palpi minute, two-segmented. Ventral plate extensive, rounded laterally, with small cervical sclerites posteriorly. Thoracic and abdominal terga divided by longitudinal median line in two parts (Figure 4A,C), thoracic pleurites small, lightly colored. Prosternum triangular, broad, precoxale small, slender. Meso- and metasternum triangular, with rounded caudal part (Figure 4F). Episterna small and lightly sclerotized, epimera inconspicuous. Legs—short, robust (Figure 4B,F). Abdominal segments A1–A8 with spiracle in upper pleurite. Tergum of A9 emarginates at the apex, with an inconspicuous midline, two dorsoapical setae, and four ventroapical setae. Segment A10 shifted to the anterior margin of A9 (Figure 4D).

#### 3.2.6. Measurements

Body length 5.3 mm (extended by the treatment in low concentration KOH; the length of the specimen when preserved in alcohol circa 4 mm), mesothorax width 0.92 mm.

#### 3.2.7. Material Examined

One specimen, Czech Republic, Moravia, Lošov (49.616° N, 17.358° E), deposited in the collection of the Biodiversity Laboratory (CATRIN-CRH).

#### 3.2.8. Note

The mature larva of *D. aurora* was reported by Kazantsev and Nikitsky [31]. Here, we describe the second instar for comparison.

#### 3.2.9. Biology

The larvae live in rotten, red-colored wood that remains wet for most of the year; typically in large stumps and trunks. Larvae were found in tree roots up to 20 cm deep in the soil. The species is locally common in mountain forests but occurs in lowlands in high-latitude regions. Larvae pupate in autumn, and adults overwinter under the bark or in wood crevices, usually in aggregations, which sometimes contain up to several dozens of individuals. They leave the wood in the early spring. Adults remain primarily inactive, either sitting on rotten wood or herb leaves under the forest canopy, close to the place where they developed. The flight of adults is slow, restricted to shaded situations and favorable weather conditions (high humidity, no wind). The adults occur in nature for about three weeks, from late April until June, depending on the local climate. The males die shortly after copulation, and the females live for another week. The females lay eggs in rotten wood, usually in shallow holes. The first instar larvae hatch after two weeks and immediately actively search for shelter deep in rotten wood. The larvae do not build tunnels and use crevices to move within the substrate. Larvae were growing rapidly in moist wood, but we did not manage to observe the whole life cycle, and the length of development remains unknown.

#### 3.2.10. Distribution

Palearctic Region: Europe from the Mediterranean to the polar circle (Lapland, Paatari Lake, 68.86° N), Northern Africa (only literature records from the Atlas Mountains), Siberia, the Kamchatka Peninsula, Primorsky Kray, Sakhalin, Korea, Japan (Hokkaido), Kuriles (Kunashir, Shikotan) [36,37,65]; Further information was taken from www.inaturalist.org, accessed on 2 July 2022. Records close to range limits were revised, and misidentifications were excluded.

#### 3.2.11. Taxonomic Decision

There are seven available species-group names in the *D. aurora* complex [56,57,58,60,61,63,64]. Kleine [37] broadened the concept of *D. aurora* and considered other names as junior subjective synonyms. Phylogenetic analyses show that this complex consists of at least two genetically distinct clades geographically separated by the Bering Strait. We consider the Nearctic clade a separate species (see Results: phylogeny). Several available synonyms are listed for *D. aurora*, and the oldest available name referring to Nearctic populations is *Omalisus coccinatus* Say, 1835. Therefore, we designate the Nearctic species as *Dictyoptera coccinata* (Say, 1835). Although we found genetic differentiation between Russian Far East and European populations, we refrain from the resurrection of *D. superba* (Motschulsky) due to the lack of high-resolution nuclear markers and information on the eventual sympatric occurrence. Similarly, the *cox1-5*′ genetic difference between sympatrically-occurring individuals of *D. aurora* from Fennoscandia (Appendix A) surpasses the widely accepted 2–3% threshold for the species delimitation [1], but see [66] for the genetic differentiation of geographically distant populations). The taxonomic status of European populations can be reconsidered when *D. aurora* is densely sampled in the whole continent.

The holotype of *O. coccinatus* is unavailable as Say’s collection was destroyed [67]. The name has been listed as a junior subjective synonym of *D. aurora* since Kleine [37], but due to an error, it is listed as a synonym of *Lygistopterus sanguineus* in the GBIF database (https://www.gbif.org/species/161597475; accessed on 1 June 2022). *L. sanguineus* does not occur in the Nearctic region, and the original description clearly states that the pronotum bears the median areola [56]. We refrain from a designation of the neotype as the present study is not a taxonomic revision, and we consider the identity of the species uncontentious [68].

## 4. Discussion

### 4.1. Morphological and Genetic Differentiation

Morphology is the traditional, cheap, and rapid source of diagnostic traits, but morphology-based identification has limitations and warrants validation by independent molecular data [2,15,65]. Our study of the very widespread morphospecies *D. aurora* reveals yet unknown genetic diversity (Figure 1) and the absence of undisputable diagnostic traits that would discriminate the genetically distant populations as candidate species (Figure 2 and Figure 3).

The intraspecific morphological variability surpasses identified differences between both species (Figure 2 and Figure 3). We found only very subtle differences between Nearctic and Palearctic specimens in the shape of the basal part and the ventral edge of the phallus (Figure 3F–Q). The students of net-winged beetles have never mentioned these. Indeed, they would not be considered sufficient to separate the Nearctic and Palearctic populations as distinct biological species if we do not have molecular data indicating genetic differentiation, reciprocal monophyly, and long-term geographical isolation. In contrast with these, male antennae (Figure 3A–D), the pronotum (Figure 2E–J), the relative strength and length of longitudinal elytral costae (Figure 2A–D), and the general shape of the male genitalia (Figure 3F–Q) are quite variable within populations. *D. aurora*’s genetic divergence supports a hypothesized causality between low vagility and high diversification rate [7,12,69]. Yet, the placement of all populations in a single morphospecies challenges the intuitively expected link between genetic and morphological diversification.

The levels of mtDNA divergence detected in the *D. aurora* clade are comparable to those often found between morphologically species pairs in other insects [1,15,66] and also between the sister species of net-winged beetles [12,27]. The genetic diversification between Nearctic and Palearctic populations of the *D. aurora* complex is higher than between many species’ pairs identified in previous studies. The split between the Nearctic and Palearctic populations was estimated to be 15.75 mya (Figure 2B). Therefore, we split the *D. aurora* complex into two (semi)cryptic species with ~16 million years of independent evolution.

The genetic divergence exceeds thresholds used for algorithmic identification based on barcode data. Eleven bins (clusters) are proposed in the BOLD database for the *D. aurora* complex based on 67 barcodes (www.boldsystem.org, accessed on 1 June 2022). It is an anomaly as most *cox1-3*′ mtDNA clusters approximately correspond with biological species, or the species are split into only a few genetically distinct groups if data originate from disjunctive areas [2,66].

Although we deal with a single morphospecies and morphological stasis is a macroevolutionary and paleontological concept [70,71], it is worth discussing the processes possibly leading to such an undisputable lack of morphological divergence (Figure 2 and Figure 3). All net-winged beetles share very similar biology, yet most closely related species are morphologically distinct [27,32,72]. For example, their distinctiveness is often a result of advergence to various Müllerian co-mimics [28,50,73,74,75]. Further diagnostic traits are regularly found in the male genitalia that plays a role in sympatric speciation [76,77]. Therefore, intrinsic constraints undoubtedly play a role in morphological stasis, but ecological factors also affect the divergence of the phenotype.

*D. aurora* and *D. coccinata* belong to a complex of red-colored net-winged beetles. This pattern is ancient [50] and widespread in the whole Holarctic region. *D. aurora* resembles *Lopheros rubens* (Gyllenhal), *E. cosnardi* (Mulsant), *B. taygetanus* (Pic), *B. longicornis* (Reiche), and *P. rubripes* (Pic) in Europe and *Dictyoptera* spp., *Helcophorus* spp., *Conderis* spp., and *Lyponia* spp. in East Asia. Similarly, *D. coccinata* belongs to the complex of red lycids in Northern America, i.e., *D. hamata* (Mannerheim), *D. simplicipes* (Mannerheim), and *Punicealis munda* (Say). Aposematic color patterns and mimetic rings persist for a long time [13,50,77]. Therefore, stabilizing selection in mimetic rings is an option for explaining the long-term phenotypic stasis in the *D. aurora* complex.

The origin of the *D. aurora* complex and its striking morphological uniformity stands in contrast with a new Ferreira’s hypothesis that stable paleoenvironment is a driver for conserved morphology in a paedomorphic lineage [44]. The hypothesis is based on a single, ~15 million years (my) old, Dominican amber neotenic species, its extant relative placed in the same genus, and the stable position of Hispaniola in the tropics. All populations of the *D. aurora* complex are morphologically uniform (Figure 2 and Figure 3), yet the morphological stasis due to stable environmental conditions is exceptionally improbable as *D. aurora* occurs from Northern African cedar forests to tundra low shrub ecosystems beyond the polar circle in Fennoscandia and *D. coccinata* from dry shrub and forest vegetation in Arizona to the polar circle area in Alaska and the humid temperate to subtropical forests of Georgia and northern Florida. Following the argumentation presented by Ferreira et al. [44], we would have to analogically propose that the unstable and diverse environments select for the morphological stasis of the *D. aurora* complex.

The linkage of morphological stasis and neoteny is another aspect of Ferreira’s theory that is contradicted by the conserved morphology of fully metamorphosed *D. aurora* (Figure 2 and Figure 3). First of all, the neotenic net-winged beetles are morphologically disparate. They represent 2% of net-winged beetle species diversity but account for 25% of the described morphology-based supergeneric taxa (12 of 48 proposed; [18,78,79]. We might also consider that the ontogenetic modifications have a substantial impact on the phenotypes, and neoteny, or more broadly pedogenesis, are important macroevolutionary drivers producing morphological disparity [80]. The net diversification rate of neotenics must be considered, i.e., speciation and extinction [81]. The high extinction rate is possibly the reason why a low net diversification was observed in neotenics after the colonization of new landmasses and an initial short-term diversification burst [82,83].

Another question is the time factor. The morphologically uniform *D. aurora* complex split from its closest, phenotypically similar relatives at least 30 my before the time of Dominican amber (54 versus 23 my or 15 my) [44,84]. An ancient origin of the *Dictyoptera*-like phenotype is also supported by Priabonian Erotinae [85,86,87]. The earliest split between the cryptic species, *D. aurora* and *D. coccinata*, is estimated at 15.75 mya (7.18–26.06 mya 95% confidence interval; Figure 1D). The Dominican leptolycine species underlying Ferreira’s hypothesis is slightly younger than the earliest split within the morphospecies earlier designated as *D. aurora*. The age of the Dominican fossil species disqualifies it from a long-term stasis evaluation. The dated phylogenies of net-winged beetles have shown that many genera with morphologically uniform species can be traced to the Paleogene [13,28,50]. As the studies were published only recently, Ferreira and collaborators [44] possibly missed them. Still, the authors did not compare the time since the establishment of the conserved morphology of their Dominican net-winged beetle with other beetles known for morphological stasis. Examples include morphological similarity of extant and early Mesozoic elateroids; mid-Cretaceous and Priabonian lycids [88,89,90,91,92,93,94] but not [95] who reported a Cretaceous tenebrionoid beetle as a lycid by error; extant and mid-Cretaceous lymexylids [96,97], jacobsonids [98], or micromalthids [99]. Growing information on Kachin amber provides further examples in various lineages [100].

The ancient morphological stasis was also recovered by dated phylogenies [101,102]. To sum up, there is plenty of evidence that morphological similarity of closely related species (within genera, usually with the earliest common ancestor hypothesized in the Paleogene) is a rule and not evidence for a causal link between conserved morphology, the neotenic development, and stable habitat. The *D. aurora* complex is one of the well-documented examples of the long-term persistence of an ancient phenotype under very diverse and unstable environmental conditions.

### 4.2. Genetic Diversity in European Populations

The Palearctic populations of *D. aurora* do not represent a genetically homogenous group, and they split into the western and eastern clades in the five gene analyses (Figure 1C). Additionally, two distinct *cox1-5*′ haplotype groups are identified in Fennoscandia (Figure 1B). As net-winged beetles depend heavily on rotten wood, we assume that the range of *D. aurora* was pushed to the south along with forest ecosystems during glacial maxima [103]. The high genetic diversity in the Fennoscandian population shows that the area only relatively recently reclaimed by forest habitats after the retreat of the continental ice sheet was populated from two long-term separated refugia, presumably located in southeastern Russia and western Europe. Similar diversification was identified in *P. nigroruber* (Figure 1H, *cox1-5*′ divergence 4.9% in Finland). Still, it stands at odds with the genetically homogenous Fennoscandian population of the other two North European net-winged beetles, *P. minuta*, and *L. sanguineus* (Figure 1F,G). The dual colonization of Fennoscandia has been reported in some plants, mosses, and rodents [103,104,105]. Our findings open the questions of a possible reproductive isolation of sympatric Fennoscandian populations (mtDNA divergence up to 6.8%; Appendix A) and the frequency with which various dispersal routes were employed during the recolonization of Northern Europe after the LGM and which species were able to retreat only to some refugia. Further research must be based on more extensive data than we currently have at our disposal. We need to densely sample the populations from the whole range including Eastern Europe and Siberia and to employ multiple nuclear markers to robustly recover the colonization routes after the retreat of the continental ice sheet.

## 5. Conclusions

The high genetic diversification within the traditionally delimited morphospecies *D. aurora* exemplifies unlinked genetic and morphological divergence. The Holarctic distribution of *D. aurora* has been unique as no other net-winged beetle species is simultaneously known from two zoogeographic regions [17]. Regardless of morphological similarity, we have to resurrect the name *D. coccinata* for Nearctic populations. As we do not have deep insight into the structure of the whole genome, we cannot robustly identify the processes that control morphological uniformity. Provisionally, we assume the role of the stabilizing selection in the Mullerian mimicry ring (besides the possible intrinsic morphological stasis). The broad ecological valence of *D. aurora* questions the recent hypothesis linking the stable environment and morphological uniformity of net-winged beetles. The present results show the potential of poorly dispersing beetles for studies on the origin of the large regional faunas and the distribution dynamics since the LGM. With growing information content, the barcode database (www.boldsystems.org, accessed on 1 June 2022) becomes a powerful tool for taxonomy. Yet, the barcodes provide only the first clue. Therefore, we separate as full species only Nearctic and Palearctic populations for which we have additional data. The robust reconstruction of dispersal routes and eventually delimitation of further intraspecific taxa will need nuclear markers and a dense, now unattainable, sampling from the whole range.

## Figures and Tables

**Figure 2 insects-13-00817-f002:**
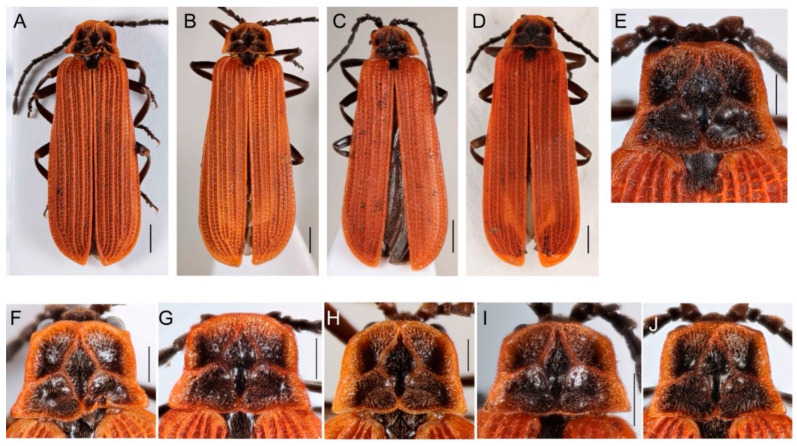
(**A**,**B**,**F**–**H**) *Dictyoptera aurora* and (**C**–**E**,**I**,**J**) *D. coccinata*. (**A**–**G**) General appearance; (**E**–**J**) Pronotum. Scales: 1 mm (**A**–**D**), 0.5 mm (**E**–**J**).

**Figure 3 insects-13-00817-f003:**
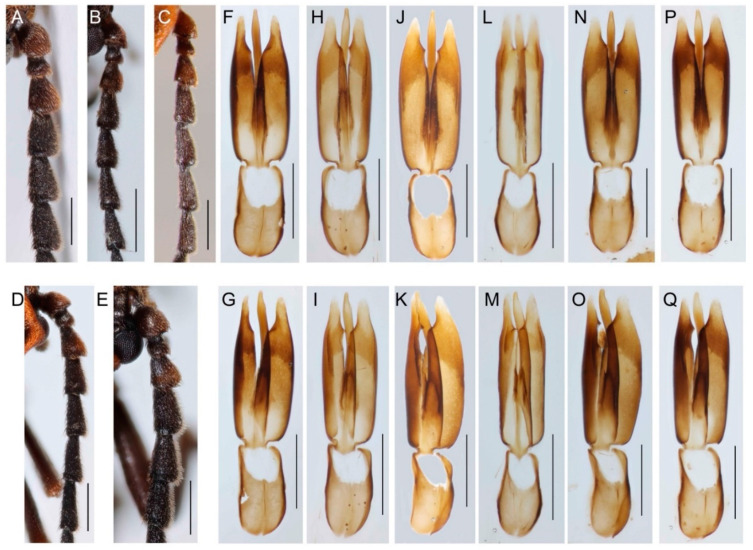
(**A**–**C**,**F**–**K**) *Dictyoptera aurora* and (**D**,**E**,**L**–**Q**) *D. coccinata*. (**A**–**E**) Antennomeres 1–6; (**F**–**Q**) Male genitalia, dorsal (upper row), and ventrolateral views (bottom row). Each column of the genitalia figures contains two views of the same individual. Scales: 0.5 mm.

**Figure 4 insects-13-00817-f004:**
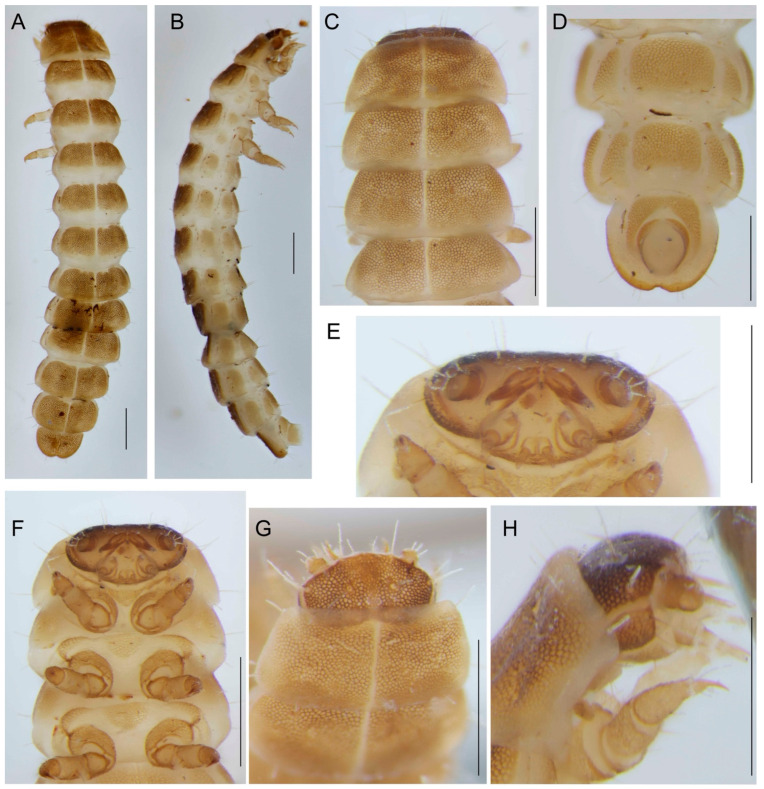
*Dictyoptera aurora* (Herbst), second instar larva, (**A**) dorsal view; (**B**) lateral view; (**C**) head, thorax, and abdominal segment 1, dorsal view; (**D**) terminal abdominal segments, ventral view; (**E**) head, frontal view; (**F**) head and thorax, ventral view; (**G**) head and pronotum, dorsal view; (**H**) ditto, lateral view. Scales 0.5 mm.

## Data Availability

The data presented in this study are available in the article.

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
