# Peer review of "Analysis of the Holarctic Dictyoptera aurora Complex (Coleoptera, Lycidae) Reveals Hidden Diversity and Geographic Structure in Müllerian Mimicry Ring"

_insects, 2022, doi:10.3390/insects13090817_

Round 1
Reviewer 1 Report
This is very nice paper. Sampling for molecular analysis is a bit low, but this is a usual problem of these studies. Fig 3 could be rearranged in a way that the top row shows elements of one species and the bottom row shows elements of another species.
Author Response
Many thanks for your review. The sampling is really a problem, and despite we tried to get further samples we had to publish with available material. D. aurora and D. coccinata are not very common, and their range covers the Holarctic region.
We considered the proposal that Fig. 3 can be rearranged, but we prefer to have rows of the genitalia of both species to enable easy comparison in the same observation angle. We expanded the legend to help the reader.
Reviewer 2 Report
1. Krivolutzkaya (2013) (line 78): not given in References (maybe 1973?)
2. Cantharis and Platycis are feminine - Cantharis sanguineus and Platycis minutus should be given as C. sanguinea and Platycis minuta.
3. B. taygetanus (line 374) should be given in italics
4. Hiekeolycus (line 413) is not yet resurrected and considered a synonym of Helcophorus - maybe it's better to use both names?
Author Response
Many thanks for the review. We have accepted all suggestions and modified the manuscript accordingly.